# The Second Highest Prevalence of Celiac Disease Worldwide: Genetic and Metabolic Insights in Southern Brazilian Mennonites

**DOI:** 10.3390/genes14051026

**Published:** 2023-04-30

**Authors:** Luana Caroline Oliveira, Amanda Coelho Dornelles, Renato Mitsunori Nisihara, Estevan Rafael Dutra Bruginski, Priscila Ianzen dos Santos, Gabriel Adelman Cipolla, Stefanie Epp Boschmann, Iara José de Messias-Reason, Francinete Ramos Campos, Maria Luiza Petzl-Erler, Angelica Beate Winter Boldt

**Affiliations:** 1Laboratory of Human Molecular Genetics, Department of Genetics, Federal University of Paraná (UFPR), Centro Politécnico, Jardim das Américas, Curitiba 81531-990, Paraná, Brazil; 2Postgraduate Program in Genetics, Department of Genetics, Federal University of Paraná (UFPR), Centro Politécnico, Jardim das Américas, Curitiba 81531-990, Paraná, Brazil; 3Laboratory of Molecular Immunopathology, Department of Clinical Pathology, Clinical Hospital, Federal University of Paraná (UFPR), Rua General Carneiro, 181 Prédio Central, 11° Andar, Alto da Glória, Curitiba 80060-240, Paraná, Brazil; 4Postgraduate Program in Pharmaceutical Sciences, Laboratory of Bioscience and Mass Spectrometry, Department of Pharmacy, Federal University of Paraná (UFPR), Av. Pref. Lothário Meissner, 632, Jardim Botânico, Curitiba 80210-170, Paraná, Brazil; 5Postgraduate Program in Internal Medicine, Federal University of Paraná (UFPR), Rua General Carneiro, 181 Prédio Central, 11° Andar, Alto da Glória, Curitiba 80060-240, Paraná, Brazil

**Keywords:** celiac disease, Mennonites, HLA-DQ2.5, HLA-DQ8, founder effect, glutathione, subdiagnosis

## Abstract

Celiac disease (CD), despite its high morbidity, is an often-underdiagnosed autoimmune enteropathy. Using a modified version of the Brazilian questionnaire of the 2013 National Health Survey, we interviewed 604 Mennonites of Frisian/Flemish origin that have been isolated for 25 generations. A subgroup of 576 participants were screened for IgA autoantibodies in serum, and 391 participants were screened for HLA-DQ2.5/DQ8 subtypes. CD seroprevalence was 1:29 (3.48%, 95% CI = 2.16–5.27%) and biopsy-confirmed CD was 1:75 (1.32%, 95% CI = 0.57–2.59%), which is superior to the highest reported global prevalence (1:100). Half (10/21) of the patients did not suspect the disease. HLA-DQ2.5/DQ8 increased CD susceptibility (OR = 12.13 [95% CI = 1.56–94.20], *p* = 0.003). The HLA-DQ2.5 carrier frequency was higher in Mennonites than in Brazilians (*p* = 7 × 10^−6^). HLA-DQ8 but not HLA-DQ2.5 carrier frequency differed among settlements (*p* = 0.007) and was higher than in Belgians, a Mennonite ancestral population (*p* = 1.8 × 10^−6^), and higher than in Euro-Brazilians (*p* = 6.5 × 10^−6^). The glutathione pathway, which prevents reactive oxygen species-causing bowel damage, was altered within the metabolic profiles of untreated CD patients. Those with lower serological positivity clustered with controls presenting close relatives with CD or rheumatoid arthritis. In conclusion, Mennonites have a high CD prevalence with a strong genetic component and altered glutathione metabolism that calls for urgent action to alleviate the burden of comorbidities due to late diagnosis.

## 1. Introduction

Celiac Disease (CD) is a chronic, immune-mediated enteropathy of the small intestine caused by exposure to dietary gluten (insoluble gliadin polypeptides found in wheat, rye, barley, and other closely related grains) in genetically predisposed individuals [1]. CD has a strong genetic association with the human leukocyte antigens (*HLA*) HLA-DQ2.5 (*DQA1*05-DQB1*02* haplotype) and HLA-DQ8 (*DQA1*03-DQB1*03:02* haplotype), which are estimated to account for up to 40% of the heritability [2]. However, only 3% of HLA-DQ2.5-positive and/or HLA-DQ8-positive Europeans consuming gluten develop the disorder, indicating that other genetic and/or environmental factors are involved in CD pathogenesis [3].

The clinical presentation varies from typical gastrointestinal symptoms such as diarrhea, steatorrhea, weight loss, bloating, flatulence, and abdominal pain to extraintestinal manifestations such as abnormal liver function, iron deficiency anemia, bone disease, and skin disorders. Affected individuals may also remain asymptomatic for a long time, despite the growing loss of their intestinal villi [4,5,6]. They are often detected through serologic testing of CD-specific autoantibodies and definitively diagnosed with a duodenal mucosal biopsy [7]. Nevertheless, clinical heterogeneity complicates the diagnostic work-up, delaying the diagnosis or allowing the disease to remain unrecognized, causing CD to be highly underdiagnosed. A gluten-free diet (GFD) is the main form of treatment. Affected individuals who are non-compliant with a GFD and/or have been lately diagnosed may develop polyautoimmunity, osteoporosis, sterility, sexual dysfunction, and dermatologic, neurological and psychiatric disorders, as well as small bowel adenocarcinoma, lymphoma, and carcinoma of the esophagus [8,9,10]. CD prevalence is higher in females, individuals with certain disorders (type 1 diabetes mellitus, autoimmune thyroiditis, Down syndrome, for example), and first-degree relatives of CD patients [11]. The global CD seroprevalence is 1.4% (95% CI = 1.1–1.7%), with the highest seroprevalence reported to date being 5.6% in the isolated Saharawi people of Western Sahara [12]. On the other hand, the global prevalence of biopsy-confirmed CD is 0.7% (95% CI = 0.5–0.9%), with the highest prevalence in Europe (0.8%) and Oceania (0.8%) and the lowest prevalence in South America (0.4%); the prevalence is 0.24% (1:417) in Curitiba, the capital city of Paraná state in Brazil [13,14]. This city also harbors three Mennonite settlements.

Mennonite is the name given to people belonging to a pacifist religious group that emerged in the 16^th^ century during the Anabaptist Movement in Europe. They have passed through at least three bottlenecks, living in isolated communities for more than 20 generations. In Brazil, approximately 1.200 Mennonites (200 families) arrived in the state of Santa Catarina in 1930. From there, several families moved to Curitiba (CTB/PR). Another 86 families moved to Rio Grande do Sul state in 1949 and founded Nova Colony (CN/Aceguá-RS). In 1951, 74 families, totaling 455 people, settled in the Witmarsum Colony (CW/Palmeira-PR) [15]. Although marriages were mostly random within the Mennonite colonies, approximately five centuries of isolation favored consanguinity [15]. Inbreeding contributes to the increase in homozygosity above the level predicted under the Hardy–Weinberg equilibrium and to the expression of specific recessive phenotypes due to allele sharing within families [16,17,18]. Thus, the Mennonites offer an excellent model for studies of complex diseases, thanks also to the accuracy of their genealogical records [19].

We performed a pilot study in January 2013 including 93 individuals from the Witmarsum Mennonite community that were previously enrolled in a genetic population study for lactose intolerance [20]. We found serologically positive anti-endomysial results in three individuals who unsuspected the disease (3.2%). This prompted us to launch an epidemiological, transversal field study lasting from October 2016 to December 2018. The daily consumption of gluten reported in the previous pilot study further justified a tracking study of celiac disease in this population. In this work, we identified one of the highest CD prevalence rates worldwide, which was associated with the distribution of predisposing HLA-DQ variants in Mennonites. Given that the prevalence of CD in first-degree relatives (FDR) of celiac patients increases by 2–21%, depending on the population, gender, and HLA-DQ genotype [21], populational screening of Mennonites is of high public health value regarding preventive medicine, allowing the identification of individuals who will benefit from early dietary intervention, i.e., a GFD.

## 2. Materials and Methods

### 2.1. Patient and Public Involvement

Representatives of the Mennonite population, concerned with the apparent high frequency of gastrointestinal symptoms and cancer in their communities, personally directed a request for the identification of the underlying causes and prevention of chronic diseases to the current coordinator of the project. This prompted us to start a survey of lactose intolerance, which was followed by the current epidemiological survey of celiac disease. We previously discussed the strategy of contact, personal interviews, and timing and location of recruitment with the leaders of all communities. We invited the population to participate through an article published in German in the local newspaper, letters, and emails distributed by the residents’ associations and through short oral communications given during the service in the local churches. Mennonite health workers (physicians, nutritionists, nurses, pharmacists, psychologists), educators, and preachers were engaged in the project from the very beginning, both to motivate participation in the study and to support the deliverance of general results (regarding findings about prevalence and prevention of celiac disease) in talks given during special meetings. All community leaders are in close contact with the project coordinator.

### 2.2. Epidemiological Study

We aimed at 200 individuals from each of the three largest settlements to reach a sample that might be representative of the Mennonite Brazilian population and with enough statistical power to identify an HLA genetic effect (higher than 80%). Inclusion criteria: Mennonite origin for at least one of the parents (sharing a common migratory route from the Netherlands to Poland, then to Ukraine, and from there again to Germany and later to Brazil or Paraguay), more than 12 years of age, capacity to understand and answer the questions of the interviews. Exclusion criteria: ceasing participation in the study.

Thirty-eight individuals who participated in the pilot study were also included in this study; they were interviewed and screened a second time. In total, we interviewed 604 Mennonites: 200 from Nova Colony (CON), a Mennonite settlement near the Uruguayan border (Aceguá-RS); 210 from Witmarsum Colony (CWI), cc. 70 Km from Curitiba (Palmeira-PR); and 194 from Curitiba (CTB), the capital city from Paraná state. To this end, we added some questions to the basic questionnaire of the National Health Survey (PNS) of 2013. Each interview lasted approximately one hour and included questions about parental ancestry, place of birth of grandparents and migratory route, eating and life habits, language, family atmosphere, exposure to mutagens, medical diagnosis of chronic diseases (confirmed by medical reports), and familial disease aggregation. Genealogical records and information given by participants were used to confirm their Mennonite ancestry. Weight, height, waist circumference, and blood pressure were measured according to the guidelines available at www.pns.icict.fiocruz.br and in the *Manual of Anthropometry* used in the 2013 PNS.

### 2.3. Serological Screening

Serum samples of 576 individuals were collected in tubes containing thixotropic gel, then centrifuged, aliquoted, and immediately stored at -20 °C. This represents nearly 95% of those interviewed in each community: 200 individuals from CON (100%); 207 from CWI (98.6%); and 169 from CTB (87.1%). Twenty-eight individuals reported having a previous result, obtained by an anti-tissue transglutaminase (tTG) or anti-deamidated gliadin-related peptide (DGP) serological test, 11 being positive. Among them, three individuals on a strict GFD were also included in the serological screening and found to be negative, as expected (Appendix A). Eight of the 11 positive cases also underwent biopsy exams that confirmed the diagnosis.

We further screened 93 individuals from CON, 80 from CWI, and 17 from CTB with a test that has a specificity of about 99% (97–100%) [22], namely indirect immunofluorescence with anti-human endomysial IgA antibody screening (IgA-EmA, Southern Biotech) on human umbilical cord cryostat sections [23]. This same test was also performed in 98 individuals from CWI and 25 from CTB that were later screened and found to be negative for tTG or DGP (see below). This double testing confirmed the diagnostic sensitivity and specificity [22]. We further performed enzyme-linked immunosorbent assay (ELISA) serological screening with tTG IgA (Euroimmun, Germany) in 27 individuals from CON, 67 from CWI, and 64 from CTB; and with DGP IgA (ORGENTEC Diagnostika Mainz, DE) in 80 individuals from CON, 58 from CWI, and 88 from CTB. None were screened for both. Both assays were performed according to the manufacturer’s instructions, using calibration curves. The results were given using positive, negative, and cut-off controls (20 units for tTG). There were no false positives. The only two tTG positive results were also IgA-EMA positive (Appendix A). Importantly, DGP-IgA is not an autoantibody but a marker of an ongoing celiac autoimmune response.

Positive results were given to the participants, or to their legal guardians, during personal visits accompanied by a Mennonite nutritionist, with instructions to adhere to a GFD. In collaboration with the physicians serving the communities, all positive cases after screening for autoantibodies (anti-DGP and/or IgA-EmA) were advised to confirm the diagnosis through duodenal biopsy, based on the histological findings of the duodenal mucosa in accordance with the Marsh classification [24]. This approach followed the European guidelines, since biopsy-sparing diagnoses are still challenging in adults, despite almost 100% sensitivity and specificity [22].

### 2.4. HLA-DQ2.5/DQ8 Genetic Screening

Blood from 391 individuals (64.7% of those interviewed) was collected in tubes containing ethylenediaminetetraacetic acid (EDTA) anticoagulant. Of those, 115 individuals were from CON (57.5%), 155 from CWI (73.8%), and 121 from CTB (62.4%). Genomic DNA was extracted from whole blood according to the Wizard^®^ Genomic DNA Purification kit protocol (Promega, Madison, WI, USA). We identified CD-predisposing *HLA* genotypes—*DQA1*05-DQB1*02* (DQ2.5) and *DQA1*03-DQB1*03:02* (DQ8)—without discriminating homozygotes from heterozygotes, by analyzing the melting curve of sequence-specific real-time PCR products using a Viia7 Real-Time PCR system (Thermo Fisher Scientific, Waltham, MA, USA). Sequence-specific primers for *DQA1*05, DQB1*02, DQA1*03,* and *DQB1*03:02* were described by others [25,26]. A fragment of the galactosylceramidase gene (*GALC*) was used as an internal amplification control. The amplification protocols and conditions differed for each CD-predisposing *HLA* allele (Appendix A).

### 2.5. Metabolomic Analysis by LC-HRMS

In this study, an untargeted liquid chromatography–high-resolution mass spectrometry (LC-HRMS) metabolomics approach was used to compare the metabolic state of healthy controls (Mennonites without CD—control group, CT) and Mennonites with CD (serologically positive, newly diagnosed in this study but not under a gluten free diet) to identify differences between their metabolite profiles. This approach was based on the concentration profile of all measurable free low molecular weight metabolites (without targeting a particular analyte/groups of analytes).

We compared five newly diagnosed Mennonites with CD (two 1:5 IgA-EMA positive, two 1:80 IgA-EMA positive, and one positive for anti-tTG IgA) with five CT (serologically negative). Sample preparation and analysis followed a modified version of an already described method [27] (Appendix B). LC-HRMS-based untargeted metabolomics analysis was carried out on an Ultra-Performance Liquid Chromatograph (UPLC) (Acquity UPLC System, Waters Corporation, Milford, MA, USA) coupled to a quadrupole time-of-flight mass spectrometer (QTOF MS) (Xevo G2-S TOF MS, Waters Corporation, Milford, MA, USA) with electrospray ionization (ESI) in positive ion mode. Chromatographic separation was performed on a Waters Acquity column HSS T3 C18 (100 × 2.1 mm, I.D. 1.8 µm). Mobile phase A was HPLC grade water containing 0.1% formic acid and mobile phase B was acetonitrile containing 0.1% formic acid. The column and the autosampler were maintained at temperatures of 25°C and 4°C, respectively. An 18 min linear gradient elution was performed as follows: 100% mobile phase A for the first 5 min, changing to 100% B over 14 min, holding at 100% B for 1 min, and finally back to 100% A at 15 min, holding for 3 min. HRMS source parameters were set with a capillary voltage of 3.2 kV in positive mode, with a cone voltage of 30 V. The cone and desolvation temperatures and gas flows (nitrogen) were of 120°C and 350°C and 20 L/h and 900 L/h, respectively. MS spectra were acquired in full scan analysis over an *m/z* range of 100–1000 Da. The data station operating software was MassLynx 4.1. A LockSpray solution (leucine enkephalin—*m*/*z* 556.2771) was used to maintain mass accuracy during the run time.

LC-HRMS analyses from the CD and CT groups resulted in one dataset (ESI^+^ mode). All LC-HRMS data (.raw files) were converted to the .mzML format using Proteowizard’s MSConvert software. After conversion, the spectra were uploaded to R 4.0.5 (Cran, https://cran.r-project.org/, accessed on 29 November 2022) and pre-processed with the MetaboAnalystR 3.0 package [28]. The feature matrix was exported to the web platform MetaboAnalyst 5.0 to perform the statistical analysis.

### 2.6. Statistical and Bioinformatic Analyses

Comparisons of quantitative variables were made with ANOVA or Kruskal–Wallis tests, depending on the normality of the distribution (tested with the Shapiro–Wilk test), using PRISM v.5.0 software (GraphPad Software, San Diego, CA, USA). We directly counted HLA-DQ2.5/DQ8 carrier frequencies in the Mennonite population and compared them with carrier frequencies calculated from the data available in the Allele Frequency Net Database (www.allelefrequencies.net, accessed on 20 August 2021) [29]. Belgians and Euro-Brazilians were the two most comparable populations with sufficient HLA haplotype and allele data to enable these comparisons (Appendix A).

In the association analysis, independence tests were performed between the variables using Fisher’s two-tailed or chi-square tests. All associations were corrected for possible confounding factors using logistic regression. When appropriate, the odds ratio was calculated with a 95% confidence interval. Two-tailed *p* values lower than 0.05 and significant after the Benjamini–Hochberg correction for multiple comparisons were considered significant. The associations found with univariate and multivariate binary logistic regression were obtained using VassarStats (VassarStats: Website for Statistical Computation; available at http://vassarstats.net, accessed on 1 March 2020) and STATA v.9.2 (StataCorp LP, College Station, TX, USA).

For the metabolomic analyses, principal component analysis (PCA), hierarchical cluster analysis (HCA), heat maps, and the metabolic pathway were evaluated using the MetaboAnalyst^®^ platform (https://www.metaboanalyst.ca/, accessed on 29 November 2022). Data filtering was performed by interquartile range (IQR), log transformed and normalized by auto scaling for the multivariate analysis. For *t*-test analysis, a *p* value < 0.05 was used. The *p* values and the t-score were used to obtain the enriched pathways through the functional analysis of the MetaboAnalyst tool [28]. The *mummichog* and gene set enrichment analysis (GSEA) algorithms were further used for pathway analyses, with a *p* value cutoff of 0.2 (only the top 10% of peaks were used).

## 3. Results

More than a half (60.1%) of the participants were female. The median age was 49 (14–89.2) years and differed between the communities for female participants, who were older in CON (*p* = 0.003). BMI values were also much higher in this last settlement, for both females and males (*p* < 1 × 10^−4^ and *p* < 4 × 10^−4^, respectively) (Table 1).

Approximately 19% of individuals reported having at least one clinically diagnosed autoimmune disease (AD), reporting type 1 diabetes, rheumatoid arthritis, autoimmune thyroiditis, and/or celiac disease. Among CD-related symptoms and diseases, joint pain was more frequent in CON (*p* = 1 × 10^−6^) and clinically diagnosed lactose intolerance was more frequent in CTB (*p* = 0.003). Among 453 individuals who were confident to answer about CD familial history, 26 (5.74%) presented affected FDRs and 15 (3.34%), affected second-degree relatives (SDRs). There were more reports of FDRs affected with celiac disease in CWI (*p* = 0.015) (Table 1).

The prevalence of CD did not differ among the investigated settlements. Eleven individuals were aware of their CD diagnosis, eight through both positive serology and biopsy exams. Ten individuals were IgA-EmA positive and newly diagnosed in our study (among those tested and found positive with other methods; there were no false positives; all were confirmed by positive IgA-EMA results). Two of them were identified as celiac in the pilot study. They were still not compliant with treatment and remained positive after a time span of almost five years. After a renewed contact during the second phase of our study, one of them adhered to a GFD and turned negative for anti-tTG (personal communication). It was not possible to evaluate the extent of intestinal epithelial injury through biopsies in these newly diagnosed Mennonite patients since most resisted to go through this invasive process. Nevertheless, given the high sensitivity and specificity of the employed serological exams, all ten individuals were considered CD patients in downstream statistical analyses. This represents 10/21 or 47.6% of undiagnosed CD cases (taking into account the whole sample, it would be 1.66%). The estimated CD seroprevalence was of 21/604 or 3.48% (95% CI = 2.16–5.27%), approximately 1:29. The prevalence of biopsy-confirmed CD was of 8/604 or 1.32% (95% CI = 0.57–2.59%), cc. 1:75.

Likely due to the high rate of undiagnosed CD patients, there was no difference between gluten consumption in the CD and non-CD groups. The median age of Mennonite CD patients was of 32 (17–69) and 64 (54–75) years for women (n = 16) and men (n = 5), respectively. Compared with non-CD female participants, female CD patients were younger (*p* = 0.015). The frequency of reported relatives diagnosed with CD differed between individuals with and without CD (*p* = 3 × 10^−5^). As expected, CD development was strongly associated with the presence of affected relatives (OR = 9.55 [95% CI = 3.28–27.75], *p* < 10^−3^). This was also evident in our study, since all six affected FDRs belonged to the group of patients who were already aware of their condition (biopsy proven). Even after the exclusion of four affected FDRs (two from one family and two from another, without excluding first-degree relatives from the non-affected group), prevalence remained high: 17/604 (2.81% or 1:36 [95% CI = 1.65–4.47%]). The distribution of non-CD ADs did not differ between the groups with and without CD (Table 1). As expected, chronic diarrhea, iron-deficiency anemia, and chronic abdominal pain were more common among CD patients, increasing between 4–7 times the odds for a CD diagnosis (Table 2).

Up to 391 individuals were screened for the main CD-predisposing HLA variants (Table 1). The frequency of HLA-DQ2.5- and/or HLA-DQ8-positive individuals was 49.7%, being 26% only HLA-DQ2.5-positive and 21.5% only HLA-DQ8-positive. Their distribution did not differ between the three investigated communities (Figure 1).

Based on data of the Allele Frequency Net Database (http://www.allelefrequencies.net/, accessed on 20 August 2021), we compared the frequencies of HLA-DQ2.5 and HLA-DQ8 carriers in the Mennonite population with those in the Belgian (715 individuals) and Euro-Brazilian (641 individuals) populations, considering both cis and trans allelic combinations (*DQA1*05* and *DQB1*02* for HLA-DQ2.5 and *DQA1*03* and *DQB1*03:02* for HLA-DQ8). The frequency of HLA-DQ2.5 carriers among the investigated Mennonites (29.67%) is almost identical to the one reported for Belgians (28.72%) but different from the one reported in Euro-Brazilians (17.49%, Fisher *p* = 7 × 10^−6^). Among 13 investigated CD patients, 12 were HLA-DQ2.5-positive. We did not identify HLA-DQ8-positive CD patients. Even so, the frequency of HLA-DQ2.5- or HLA-DQ8-positive individuals was lower in healthy individuals (49.74% vs. 92.31%). This status was associated with an increased susceptibility to CD (OR = 12.13 [95% CI = 1.56–94.20], *p* = 0.003). However, the frequency of HLA-DQ8 carriers within the Mennonite population (24.81%) was higher than in Belgians (14.07%, Fisher *p* = 1.8.x10^−6^). It was also higher in Mennonites compared to Euro-Brazilians (16%, Fisher *p* = 6.5 × 10^−6^).

Based on the metabolomic exploratory analysis, both PCA and HCA separated CD Mennonite patients without GFD from CTs; however, two low 1:5 IgA-EMA positive samples clustered with a control having close relatives with rheumatoid arthritis, whereas one control with a close relative with CD clustered with highly positive CD patients with 1:80 IgA-EMA values (Figure 2A,B). PCs 1 and 2 explain 21.6% and 15.7% of the total variance, respectively. The concentration of 31 metabolic features differed in CD patients (*p* < 0,05) (Figure 2C). Five metabolic pathways were altered: glutathione metabolism; glycosphingolipid metabolism; alanine metabolism; aspartate metabolism; and vitamin A (retinol) metabolism. The glutathione pathway was significantly altered using both *mummichog* and GSEA approaches (Figure 2D).

## 4. Discussion

Due to a peculiar demographic history, the genomic architecture of the Mennonite population has suffered the effects of genetic drift, which has altered the frequency of rare Mendelian, as well as chronic diseases [15,19]. Indeed, cc. 18% of the participating individuals in our study reported diagnosis of at least one AD, such as type 1 diabetes, rheumatoid arthritis, autoimmune thyroiditis, and/or celiac disease. This prevalence (17.9%) is in clear contrast with the worldwide AD prevalence of less than 5% [30]. Furthermore, the share of predisposing genetic variants within families is probably much higher than in the general population, increasing the prevalence of these diseases. More importantly, the numbers of reported FDRs and SDRs affected by CD allowed us to get an impression about previous knowledge of the disease in these rather closed communities (of 604 participants, 453 had previous knowledge about the disease in their families, e.g., 75%).

The CD seroprevalence among Southern Brazilian Mennonites was impressively high: 3.47% (approximately 1:29). Among all the populations analyzed, only the isolated Saharawi people of Western Sahara have a higher CD prevalence (5.6%) [12]. It is much higher than the 1.22% seroprevalence reported in the Dutch population [31], which is thought to be one of the main contributors to the current genetic pool of the Mennonites, together with Germans (1.57% seroprevalence) and Belgians (0.86%) [32,33]. Even if taking into account only biopsy-confirmed CD, the 1.32% frequency in Mennonites contrasts with the global prevalence of the disease, which ranges from 0.5% (1:200) to 1.0% (1:100) [34]. More specifically, CD seroprevalence was of 1.4%, and biopsy-based CD diagnosis was of 0.7% in a recent meta-analysis, ranging from 0.4% in South America to 0.8% in Europe and Oceania [13] and reaching 0.24% (1:417) in Curitiba [14] and 0.35% (1:286) in São Paulo [35], two Brazilian cities, and 0.37% in Germany [36]. The Mennonite CD prevalence is actually close to that in Finland (2.13%), which is also genetically isolated [37]. It should be noted that biopsy-proven CD prevalence among Mennonites was calculated based on those eight individuals that were willing to take the biopsy. Thus, it may actually be higher.

Ten individuals were identified as celiac by our serological screening, representing 47.6% of undiagnosed cases (almost 1:2). Worldwide, the proportion of diagnosed and undiagnosed cases varies from one country to another, spanning from 1:2 in Finland [38] to 1:20 in Argentina [39]. The failure to detect the disease, coupled with failure to treat it, may lead to severe comorbidities such as osteoporosis, sterility, neurological and psychiatric disorders, small bowel adenocarcinoma, lymphoma, and carcinoma of the esophagus, as well as increased morbidity [4,7,8,9,10]. Two individuals previously identified as celiac in the pilot study remained positive in the second serological screening, indicating non-compliance with the gluten-free diet. The reason for non-compliance may be related to sociocultural aspects of the Mennonite community, whose eating habits rely heavily on wheat consumption.

Through our questionnaire, we identified predisposed individuals who may benefit from early intervention, such as those with iron-deficiency anemia, weight loss, and chronic abdominal pain [40]. Approximately 9% of individuals reported a family history of CD, 5.79% for FDRs and 3.25% for SDRs. This agrees with the literature: prevalence of CD in FDRs of celiac patients increases between 2–21% depending on gender and *HLA*-*DQ* genotype and the association with other ADs is frequent in both patients and relatives [41,42,43,44]. Through a meta-analysis, others estimated a CD prevalence of 7.6% among FDRs and 2.3% among SDRs [45].

Worldwide, *HLA* typing has a high negative predictive value for the diagnosis of CD, since at least one of the susceptibility variants, either HLA-DQ2.5 or HLA-DQ8, occurs in 86–93% of patients with CD [2,46]. Interestingly, the frequency of HLA-DQ8 in Mennonites differed from the one reported for the Belgian population—which significantly contributed to the current Mennonite gene variant pool. This is most probably the result of a founder effect. Moreover, more than half of HLA-DQ2.5/DQ8 Mennonite carriers do not have CD (50.3%, compared with 40% among Europeans without CD) [3,43]. These individuals may also bear protective genetic/epigenetic/microbiota variants that are yet to be discovered.

Furthermore, HLA-DQ2.5 and HLA-DQ8 frequencies were significantly higher in Mennonites than in Euro-Brazilians. This is a result of prime epidemiological relevance for the design of public health policies, indicating a higher general susceptibility to the disease than most Brazilians with European ancestry.

The HLA-DQ2.5 carrier frequency among Mennonite patients (92.31%) was higher than that reported in 102 Turkish pediatric patients (76%) [47]. As in our study, none of the Turkish patients were HLA-DQ8 positive. One of our CD patients was neither HLA-DQ2.5 nor HLA-DQ8 positive, indicating the influence of yet unknown predisposing genetic variants in HLA-DQ2.5/DQ8-negative CD Mennonite patients. The absence of HLA-DQ2 or HLA-DQ8 was also formerly observed by others in Brazilian patients [48]. In fact, several non-*HLA* genes, whose products influence the integrity of the intestinal barrier and the quality of the immune response, modulate the susceptibility to the disease [34]. About 15% of the genetic susceptibility to CD is explained by 57 single nucleotide polymorphisms located in 39 non-*HLA* loci, identified through genome-wide association studies [49,50]. Furthermore, HLA-DQ2.5/DQ8 contribution to CD in monozygotic twins only ranges from 25 to 40%. Thus, it is important to not exclude strongly suspicious cases from diagnostic workup based solely on HLA-DQ2.5/DQ8 negativity.

Interestingly, low IgA-EMA-positive individuals presented a metabolic profile similar to controls with close relatives with either CD or rheumatoid arthritis, diseases that share pathological pathways [51]. In all patients, the glutathione pathway was consistently altered using both algorithms (*mummichog* and GSEA) in the pilot metabolomic investigation. This pathway acts in the antioxidant defense and turns endogenous and exogenous compounds more soluble and non-toxic [52,53]. Indeed, glutathione was reduced in CD patients compared with controls. The levels of cysteine, the main amino acid in the synthesis of glutathione, were also reduced in children with CD before a GFD [54,55,56]. Furthermore, a high concentration of reactive oxygen species is associated with bowel damage (Marsh score). This corroborates the relationship between celiac disease and the glutathione pathway, one of the main processes to inhibit the damage caused by ROS [57].

Our study has limitations. First, we could not confirm the positive serological IgA-EMA results with positive biopsies. However, given the high specificity of IgA-EMA testing, false positives are rather unexpected. Second, IgA-deficient individuals may appear negative in our serological screening. Despite being uncommon, they may occur more often among CD patients. Missed CD individuals, in this case, would have increased the prevalence values that were already very high. Third, we did not genotype individuals for HLA-DQ2.2, which also contributes to CD susceptibility. In a next phase of our genetic–epidemiological investigation, we will screen for other HLA alleles and also for IgA deficiency in the Mennonite population as recommended [22].

Finally, our results indicate that roughly one in thirty Mennonites are expected to present with CD. Of those affected, half are unaware of the disease. Most are HLA-DQ2.5-positive, but a role for HLA-DQ8, whose frequency is cc. 1.8 times higher than in original contributing populations, cannot be excluded because the CD patient sample is small. Both the HLA-DQ2.5 and HLA-DQ8 carrier frequencies are cc. 1.6 times higher in Mennonites than in Euro-Brazilians, highlighting the epidemiological relevance of this work for public health policies. The glutathione pathway seems more affected in Mennonites with CD before compliance to a GFD. Further identification of other associated genetic, epigenetic, and environmental factors shall help define the CD risk of this population and lay the foundations of future strategies of preventive medicine.

## Figures and Tables

**Figure 1 genes-14-01026-f001:**
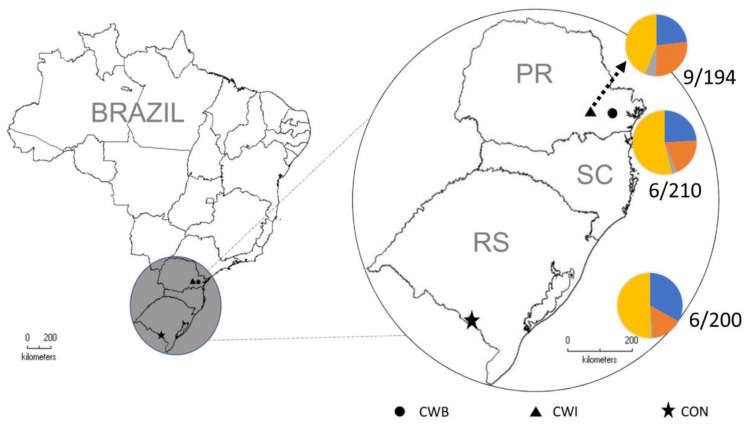
Frequencies of positive serological IgA-EMA results and of HLA-DQ2.5 and DQ8 carriers in each of the investigated Southern Brazilian Mennonite communities (HLA-DQ2.5^-^ DQ8^-^ in yellow, HLA-DQ2.5^-^ DQ8^+^ in red, HLA-DQ2.5^+^ DQ8^-^ in blue, HLA-DQ2.5^+^ DQ8^+^ in gray). PR = Paraná State, SC = Santa Catarina State, RS = Rio Grande do Sul State, CON = Nova Colony, CWI = Witmarsum Colony, CTB = Curitiba.

**Figure 2 genes-14-01026-f002:**
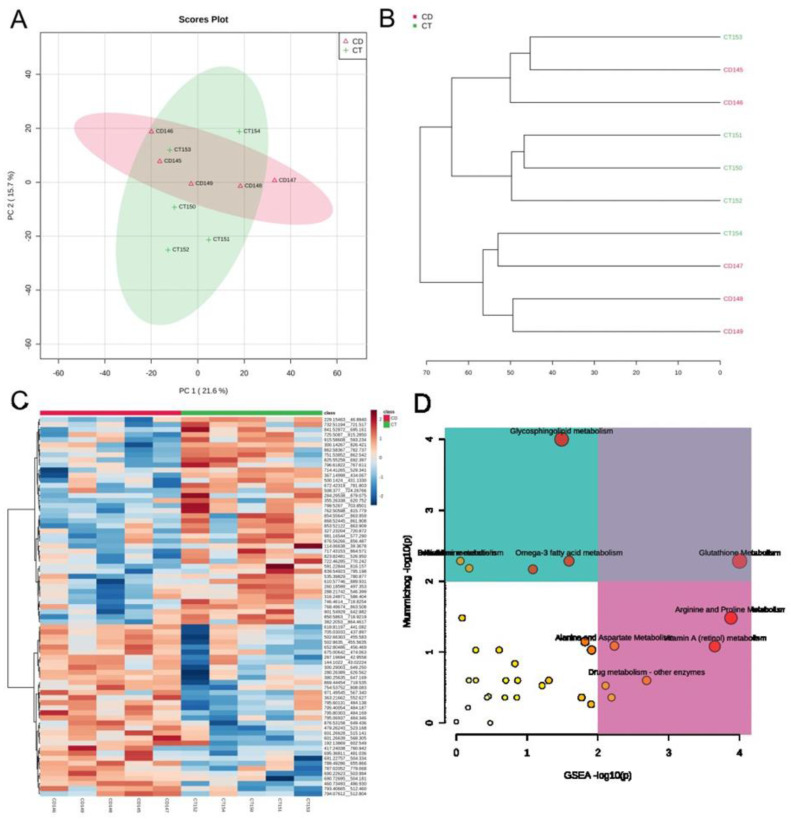
Exploratory metabolomics in the sera of CD Mennonites without a GFD. (**A**) Principal component analysis of serum metabolites identified in celiac patients (CD) and controls (CT). DC146 and DC145 had a 1:5 EMA positivity and clustered together with CT153, who is unaffected but has close relatives with rheumatoid arthritis. CT154 did not group with the others using this approach. Despite being serologically negative, this individual has close relatives with CD. (**B**) Hierarchical clustering of metabolic CD and CT profiles. Here, DC146, DC145, and CT153 also grouped together, and CT154 grouped with CD patients presenting high IgA-EMA positivity (1:80). (**C**) Heat map analysis of hierarchical clustering of CD and CT individuals. In blue: decreased metabolite levels; in red: increased metabolite levels. (**D**) Scatter plots of pathway enrichment analysis provided by *mummichog* and its integration with GSEA. The color and size of each circle correspond to its *p* value and enrichment factor, respectively. Darker tones indicate more statistically relevant predicted pathways. The size of each dot represents the ratio between significant pathway hits and the expected number of compound hits within the pathways. The glutathione pathway (upper square, right) was the only pathway found to be significant using both algorithms.

**Table 1 genes-14-01026-t001:** Distribution of demographic characteristics, symptoms, diagnoses, and HLA subtypes in the Mennonite population samples.

	CON	CWI	CTB	*p* Value	Total	CD Patients	Non-CD Patients	*p* Value
**N**	N = 200	N = 210	N = 194		N = 604	N = 21	N = 583	
** Gender (F/M)**	120/80	136/74	107/87	0.143	363/241	16/05	347/236	0.173
	**Median** **(min-max)**	**Median** **(min-max)**	**Median** **(min-max)**		**Median** **(min-max)**	**Median** **(min-max)**	**Median** **(min-max)**	
** Age–female (years)**	57.0 (14.0–83.0)	46.0 (14.4–89.0)	44.2 (14.0–89.2)	0.003 q = 0.026	49.0 (14.0–89.2)	32.0 (17.1–69.0)	50.0 (14.0–89.2)	0.015 q = 0.039
** Age–male (years)**	54.0 (17.0–82.0)	53.7 (12.0–92.1)	49.5 (15.9–88.0)	0.241	51.0 (12.0–92.1)	64.6 (54.9–75.0)	51.0 (12.0–92.1)	0.095
** BMI–female**	27.6 (15.8–45.8)	24.6 (16.2–35.2)	23.6 (15.1–40.6)	1 × 10^−4^ q = 0.013	25.2 (15.1–45.8)	22.4 (19.6–45.8)	25.5 (15.1–44.5)	0.235
** ** **BMI–male**	28.4 (19.2–42.9)	26.2 (16.1–47.6)	26.3 (16.6–38.7)	4 × 10^−4^ q = 0.015	27.1 (16.1–47.6)	27.4 (26.0–28.9)	26.9 (16.1–47.6)	0.724
	**% (n/total N)**	**% (n/total N)**	**% (n/total N)**		**% (n/total N)**	**% (n/total N)**	**% (n/total N)**	
**Symptoms/diagnoses**								
** Chronic diarrhea**	2.11 (4/190)	7.14 (13/182)	5.70 (9/158)	0.068	4.90 (26/530)	19.05 (4/21)	4.32 (22/509)	0.015 q = 0.010
** Weight loss**	3.68 (7/190)	7.94 (10/126)	5.06 (8/158)	0.251	5.27 (25/474)	15.00 (3/20)	4.85 (22/454)	0.081
** Iron-deficiency anemia**	7.37 (14/190)	10.94 (14/128)	8.23 (13/158)	0.527	8.61 (41/476)	38.09 (8/21)	7.25 (33/455)	1 × 10^−4^ q = 0.018
** Bloating**	31.58 (60/190)	28.96 (53/183)	27.21 (43/158)	0.663	29.37 (156/531)	38.09 (8/21)	29.02 (148/510)	0.462
** Constipation**	12.63 (24/190)	19.04 (24/126)	20.88 (33/158)	0.099	17.09 (81/474)	10.00 (2/20)	17.40 (79/454)	0.549
** Chronic abdominal pain**	16.32 (31/190)	16.48 (30/182)	15.28 (24/157)	0.951	16.07 (85/529)	42.85 (9/21)	14.96 (76/508)	0.002 q = 0.021
** Joint pain**	26.84 (51/190)	9.52 (12/126)	8.33 (13/156)	1 × 10^−6^ q = 0.002	16.10 (76/472)	5.00 (1/20)	16.59 (75/452)	0.223
** Diabetes mellitus type 1**	1.06 (2/188)	0.79 (1/126)	2.46 (3/122)	n.c.	1.38 (6/436)	0 (0/19)	1.44 (6/417)	nc
** Irritable bowel syndrome**	5.82 (11/189)	5.00 (6/120)	4.04 (4/99)	0.887	4.90 (20/408)	6.25 (1/16)	5.10 (20/392)	1
** Lactose intolerance**	1.62 (3/185)	3.35 (6/179)	9.59 (14/146)	0.003 q = 0.028	4.51 (23/510)	4.76 (1/21)	4.50 (22/489)	1
** Autoimmune thyroiditis**	10.47 (20/191)	11.29 (14/124)	14.74 (23/156)	0.454	12.10 (57/471)	15.00 (3/20)	12.36 (56/453)	0.727
** Rheumatoid arthritis**	11.30 (20/177)	4.00 (5/125)	5.52 (9/163)	0.031 q = 0.047	7.31 (34/465)	0 (0/19)	7.62 (34/446)	nc
** Celiac disease**	3.00 (6/200)	2.86 (6/210)	4.64 (9/194)	0.560	3.48 (21/604)	-	-	
** Any autoimmune disease ***	20.90 (37/177)	15.08 (19/126)	20.24 (33/163)	0.400	19.10 (89/466)	15.79 (3/19) *	19.24 (86/447) *	1
**Familiar aggregation**								
** FDR with CD**	3.21 (6/187)	11.21 (12/107)	5.03 (8/159)	0.015 q = 0.044	5.74 (26/453)	31.58 (6/19)	4.61 (20/434)	3 × 10^−4^ q = 0.015
** SDR with CD**	1.60 (3/187)	4.81 (5/104)	4.43 (7/158)	0.221	3.34 (15/449)	21.05 (4/19)	2.56 (11/430)	0.002 q = 0.023
** Any relative with CD**	5.88 (11/187)	14.95 (16/107)	6.29 (10/159)	0.013 q = 0.036	8.17 (37/453)	36.84 (7/19)	4.61 (20/434)	3 × 10^−5^ q = 0.010
**HLA**	**N = 115**	**N = 155**	**N = 121**		**N = 391**	**N = 13**	**N = 378**	
** DQ2.5 (+)**	33.91 (39)	29.03 (45)	26.45 (32)	0.559	29.67 (116)	92.31 (12)	27.51 (104)	3 × 10^−6^ q = 0.007
** Only DQ2.5 (+)**	33.04 (38)	23.23 (36)	23.97 (29)	0.159	26.09 (102)	92.31 (12)	24.07 (90)	1 × 10^−6^ q = 0.005
** DQ8 (+)**	16.52 (19)	32.90 (51)	22.31 (27)	0.007 q = 0.034	24.81 (97)	0	25.66 (97)	0.044 q = 0.05
** Only DQ8 (+)**	15.65 (18)	27.10 (42)	19.83 (24)	0.072	21.48 (84)	0	22.22 (84)	0.080
** DQ2.5 (+) and/or DQ8 (+)**	49.57 (57)	56.13 (87)	46.28 (56)	0.248	51.15 (200)	92.31 (12)	49.74 (188)	0.003 q = 0.031

*p* values were calculated through the chi-square test for most variables, except for BMI and age distributions (non-parametric Mann–Whitney or Kruskal–Wallis tests), gender, type 1 diabetes mellitus, and first- and second-degree relatives (Fisher exact t-test). q = correction with the Benjamini–Hochberg method for multiple comparisons. * autoimmune disease (AD): type 1 diabetes, arthritis, and/or autoimmune thyroiditis, excluding CD for the analysis in this group. Age is in years. BMI = body mass index, CD = celiac disease, CON = Nova Colony, CWI = Witmarsum Colony, CTB = Curitiba, F = female, FDR = first-degree relative, M = male, *p* = *p* value, SDR = second-degree relative, n = number of individuals with the evaluated characteristic, N = total number of individuals.

**Table 2 genes-14-01026-t002:** Common symptoms associated with CD. CD = celiac disease, Non-CD = non-celiac disease, n = number of individuals with the evaluated characteristic, N = total number of individuals, CI = 95% confidence interval, OR = odds ratio, *p* value = probability of significance. All associations were corrected for possible confounding factors (gender and age) using logistic regression.

Symptoms	CD% (n/N)	Non-CD% (n/N)	OR	95% CI	*p* Value
**Chronic diarrhea**	19.05 (4/21)	4.32 (22/509)	5.20	1.61–16.78	0.006
**Iron-deficiency anemia**	38.09 (8/21)	7.25 (33/455)	7.86	3.04–20.33	<10^−3^
**Chronic abdominal pain**	42.85 (9/21)	14.96 (76/508)	4.26	1.73–10.46	0.002

## Data Availability

The data presented in this study are available on request from the corresponding author. The data are not publicly available due to privacy restrictions given by the General Law of Data Protection (LGPD) in Brazil.

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
