# Peer review of "The Second Highest Prevalence of Celiac Disease Worldwide: Genetic and Metabolic Insights in Southern Brazilian Mennonites"

_genes, 2023, doi:10.3390/genes14051026_

Round 1

Reviewer 1 Report

The authors, Oliveira, et al. present a celiac disease prevalence study comprising interviews, serologic, histologic and genetic assessments as well as metabolomic analysis.

I have the following comments regarding the manuscript:

1.     Abstract part: line 27 and line 28: please rephrase to let the reader know that not all the 604 interviewed Mennonites were also screened for autoantibodies and/or genotyped as we later find in the text.

2.     Methods and results part:

line 123: “we interviewed 684 Mennonites” – please provide a clarification: 604 versus 684 are overlapping numbers, or different cohorts?

line 151-160 includes a description of the methodology applied to obtain the sero-prevalence of the CD in the studied population. Please provide clearer information on the results:

-       How many of the 576 serology screened individuals were positive for tTG

-       How many of the 576-serology screened individual were positive for DGP - IgA

-       If not all 576 individuals were screened for both (tTG and DGP) please provide also this information

-       Please remind the reader that DGP-IgA is not an autoantibody, not part of the autoimmune markers of CD

3.     How many of the tTG and DGP-IgA screened individuals (all positives in numbers for each of the tested antibodies) respectively were further assessed for the presence of the EMA. How many of the tTG positive/ DGP positive were also EMA positive. Final results of 10 EMA positive individuals derived from how many tTG and DGP positive individuals respectively?

line 250-254. When collecting the given information 576 was the total number of serological screened individuals. We do not have info on the 28 individuals not screened. Please provide clear info on this and recalculate the seroprevalence.  Line 237 ( Table) – 583 non-Cd patients and 21 CD patients (21+583 = 604). But the serology was tested in only 576 individuals.

The biopsy proven CD prevalence is calculated using the 8 individuals willing to take the biopsy. Please highlight this in the discussion. Now it is just indirectly understandable from different parts of the manus.

 I do not see the point of the 10/21 calculation and the use of this in the results and discussions (line 254 and line 330 respectively). The 8+3 earlier detected were they also detected during the screening or were they clinically suspected? Please provide info.

A flow chart of the seroprevalence study including all the info above might be useful for a better understanding of the results. The flow chart could be presented as a supplementary material.

4.     Discussion part: please provide info and comment on the reagent kit used to determine the tTG serology – the results were given using a calibration curve or one positive control only? Please provide median and quartile for the results of the tTG serological results. Please comment on the possible false positive results of the serology. How many of the tTG positive/ DGP positive were also EMA positive. 

Author Response

Reviewer 1:

The authors, Oliveira, et al. present a celiac disease prevalence study comprising interviews, serologic, histologic and genetic assessments as well as metabolomic analysis.

I have the following comments regarding the manuscript:

  1. Abstract part: line 27 and line 28: please rephrase to let the reader know that not all the 604 interviewed Mennonites were also screened for autoantibodies and/or genotyped as we later find in the text.
  2. Done as requested.
  3. Methods and results part:

line 123: “we interviewed 684 Mennonites” – please provide a clarification: 604 versus 684 are overlapping numbers, or different cohorts?

  1. We corrected the number in section 2.2, for this study it is actually 604 (see Table 1).

line 151-160 includes a description of the methodology applied to obtain the sero-prevalence of the CD in the studied population. Please provide clearer information on the results:

R.: We screened 93 individuals from CON; 80 from CWI; and 17 from CTB with a test that has a specificity of about 99% (97-100%) [22], namely indirect immu-nofluorescence with anti-human endomysial IgA antibody screening (IgA-EmA, Southern Biotech) on human umbilical cord cryostat sections [23]. This same test was also done in 98 individuals from CWI, and 25 from CTB, later screened and found neg-ative for tTG or DGP (see below). This double testing confirmed diagnostic sensitivity and specificity [22]. We further performed enzyme-linked immunosorbent assay (ELISA) serological screening with tTG IgA (Euroimmun, Germany) in 27 individuals from CON; 67 from CWI; and 64 from CTB; and DGP IgA (ORGENTEC Diagnostika Mainz, DE) in 80 individuals from CON; 58 from CWI; and 88 from CTB. None were screened for both. Both assays were performed according to the manufacturer’s in-structions, using calibration curves. The results were given using positive, negative and cut-off controls (20 units for tTG). There were no false positives. The only two tTG positive results were also EMA positive (supplementary Table 1). This information was included as a table in the supplementary material and inserted in section 2.3 (“Serological screening”).

Supplementary Table 1 – Serological screening in the Mennonite population.

Test

CON +

CON -

N

CWI +

CWI -

N

CTB +

CTB -

N

N +

N -

Total

DGP

0

80

80

0

58

58

0

88

88

0

226

226

tTG

1

26

27

0

67

67

1

63

64

2

156

158

EMA

6

87

93

4

78

82

0

17

17

10

184

192

Previous

0

0

0

3*

0

3*

6

19

25

9

17

28

Total

7

193

200

7

203

210

7

187

194

21

583

604

DGP - anti-deamidated gliadin-related peptide, tTG - anti-tissue transglutaminase, EMA - human endomysial IgA antibody screening, CON – Colônia Nova, CWI – Colônia Witmarsum, CTB – Curitiba, N – number of individuals. * screened as EMA negative, but had been previously diagnosed for CD and were already in GFD, at the time of participant recruitment.

-       How many of the 576 serology screened individuals were positive for tTG (595)?

R: Two were positive. These details were included in supplementary Table 1.

-       How many of the 576-serology screened individual were positive for DGP – IgA

R.: None was positive.

  1. How many of the tTG and DGP-IgA screened individuals (all positives in numbers for each of the tested antibodies) respectively were further assessed for the presence of the EMA. How many of the tTG positive/ DGP positive were also EMA positive. Final results of 10 EMA positive individuals derived from how many tTG and DGP positive individuals respectively?

R.: The screening started with EMA and continued with tTG and DGP. Only two individuals were found positive with tTG, none with DGP, and these individuals were also EMA positive. Thus, EMA positive individuals consisted of 8 EMA positive and two double-positive tTG / EMA individuals. For further details, see supplementary Table 1.

line 250-254. When collecting the given information 576 was the total number of serological screened individuals. We do not have info on the 28 individuals not screened. Please provide clear info on this and recalculate the seroprevalence.  Line 237 ( Table) – 583 non-Cd patients and 21 CD patients (21+583 = 604). But the serology was tested in only 576 individuals.

R.: Prevalence cannot be calculated simply by positive serology, due to GFD (GFD ceases autoantibody production). At least three individuals who reported previous CD diagnosis, initiated GFD before recruitment and were EMA negative in the serological screening. All 28 individuals not screened in our study had been recently tested for CD before by tTG or DGP tests, 11 being positive. Eight of 11 also underwent biopsy exams and confirmed the diagnosis. This information was added to section 2.3 in the manuscript, as required.

The biopsy proven CD prevalence is calculated using the 8 individuals willing to take the biopsy. Please highlight this in the discussion. Now it is just indirectly understandable from different parts of the manuscript.

R.: Done as required, at the end of the second paragraph of the discussion.

 I do not see the point of the 10/21 calculation and the use of this in the results and discussions (line 254 and line 330 respectively). The 8+3 earlier detected were they also detected during the screening or were they clinically suspected? Please provide info.

R.: They were detected earlier, with confirmed diagnosis. Among them, we included three in our serological screening, and found them to be negative (since they already initiated a GFD). This information was included in section 2.3.

A flow chart of the seroprevalence study including all the info above might be useful for a better understanding of the results. The flow chart could be presented as a supplementary material.

R.: Instead of a flowchart, we decided to include this extra information in a supplementary table, as formerly mentioned in our previous answers.

  1. Discussion part: please provide info and comment on the reagent kit used to determine the tTG serology – the results were given using a calibration curve or one positive control only? Please provide median and quartile for the results of the tTG serological results. Please comment on the possible false positive results of the serology. How many of the tTG positive/ DGP positive were also EMA positive. We are looking forward to hearing from you soon.

R.: We used a calibration curve for tTG serology. The test was performed according to the manufacturer's instructions, taking into account the cutoff of 20 units (without calculating median and IQR). There were no false positives. The only two tTG positive results were also EMA positive. We included this information in Material and Methods (setion 2.3) and Results (3rd paragraph).

Reviewer 2 Report

Title:

Second highest prevalence of celiac disease worldwide: genetic and metabolic insights in South Brazilian Mennonites

Review:

Author has performed an interesting research study on the Mennonites population that has been isolated for 25 generations. Author has performed HLA DQ2.5 and DQ8 genotyping along with metabolic profiling approach elucidating high prevalence and altered glutathione pathway. However, there are some serious corrections that need to be rectify in the manuscript.

Major corrections

Introduction

1)    Instead of stating fraction of HLA DQ2.5-positive and/or HLA DQ8 individual, it is advisable to write the proper stats with references

2)    Line 68-69; There is no need to write (reviewed by)

3)    Page 2, paragraph 3, Line 75-83 must be rewrite; Instead of providing the history of Mennonite population, author is advised to write the genetic background and disease prevalence in relation to celiac disease; number of genetic and epidemiological studies conducted and most important findings in short.

4)    Page 2, paragraph 3, Line 82-92; Paragraph seems incomplete. It is advisable to re write this part too depicting the novel objective of the study and the expected outcomes at the end

Materials and Methods

1)    There are discrepancies in the sample size in more than one section of the manuscript, which arise confusion throughout the study. Author is advised to make a flow chart depicting the number of cases from the enrollment to the number of evaluation of the cases in the study.

2)    Line 121-123; If the 38 individuals from the pilot study were screened again, then the total number of sample size should be 684+38.

Here author needs to specify clearly whether these 38 individuals were only screened for serology (anti-tTG-IgA or EMA) or interviewed also. And also these individuals were participated in the total sample size or not.

3)    Author needs to describe the inclusions and exclusions of the study precisely in the manuscript with the basis of proposed sample size

4)    There is no need to provide the irrelevant information in the manuscript. Information related to the current study must provide in the manuscript

a)     Line 98-99; Lactose intolerance survey

b)    Line 104-105; article in local Mennonite newspaper

5)    Line 117-119 repeated second time. It is advisable to shorten the sentence if it contains important information.

6)    Line 120-121; Reason to conduct the study and the study type must come under the objective section, not in the method section.

7)    Line 141-144; Gender, Age and BMI information must come in results section not in method section.

8)    Figure 1: It is advised to the author to create a chart depiction number of cases enrolled from the individual communities, number of serology positive cases from the enrolled cases and the percent of HLA DQ2.5 and DQ8 genotypes.

9)    Line 147: Serology analysis performed 576 cases. Author has interviewed 684 cases (with 38 cases from pilot study). Author needs to specify what happened to the rest of the cases.

10) Line 159-162; Author has mentioned that the serology positive cases were advised to duodenum biopsy based diagnosis of celiac disease, reference 20

Here author needs to specify which diagnostic guidelines were followed for the diagnosis of celiac disease.

11) Line 165; blood from 400 cases was collected for HLA genotyping. Was there a specific reason to genotype the lesser number of cases? Why the rest cases were left for genotyping?

12) Author performed HLA DQ2.5 and DQ8 genotyping.

Is there any specific reason author did not genotyped HLA DQ2.2 region? Although lesser number of cases show susceptibility with HLA DQ2.2 genotype and has less frequency in celiac disease patients, but still this region has its own importance and it is very important to perform the genetic analysis of this region too. Also, along with DQ2.5, it completes the HLA DQ2 genotype. Author needs to provide very specific reason for that.

13) Line 179; author described that “we compared 5 CD without GFD”

This statement creating a confusion and require a clarification whether these cases:

a)     Newly diagnosed Celiac disease cases

b)    Cases from earlier performed pilot study or the cases that were serology positive but did not consuming gluten free diet

c)     Cases non-compliant to Celiac disease

14) Line 180-181; author need to write the sample type and sample preparation in short

15) In the metabolomic analysis section, author needs to write the pathway and the groups of analytes author is targeting and why?

16) No need of table 1 and 2 in supplementary sheet

Results

1)     In the table 1, author needs to write the whether the age is in months or years

2)    Did author calculated the frequency of the FDRs and SDRs from the total number of celiac disease cases or from the total number of the cases screened? And why?

3)    If all the 10 cases did not underwent biopsy procedures then what guidelines were used to make the diagnosis of celiac disease and how? Line 253

4)    Author has previously mentioned that 684 cases were interviewed and out of which 576 samples were tested for serology then why author has calculated the prevalence (serology based and biopsy proven) from 604 cases? From where this 604 figure turn out? Line 255-257

5)     Why author calculated the frequency of undiagnosed celiac cases from 21 cases and not from the total sample size?

6)    Unsuspected celiac disease patients. Line 258

Unsuspected is not a standard term, either use atypical or any other specific/appropriate term

7)    In the method section author mention 400 cases were HLA genotyped but here it is written that 391 cases were screened for the HLA variants? Clarify

8)    Line 282- provide the frequency of HLA genotypes in controls too

9)    Previously 21 celiac disease cases were described but in line 290 they are 13 only. Clarify

10) Figure 1 in supplementary sheet must be included in the main text of manuscript

Discussion

1)    How did author calculate frequency of underdiagnosed celiac disease cases as 47.6%. why it is not calculated from the total sample size?

2)    Please provide the ratio from the current study as depicted from Finland and Argentina in line 331-332

3)    Author did not perform the analysis of HLA DQ2.2 region, a well-known predisposing genotype, and stating the possibilities of unknown predisposing genetic variants. Why? Provide hard reasons to support the statement

4)    Line 377; please specify the both algorithms in pilot metabolomic investigation

Minor but important corrections

Author needs to write the full forms of the following at once in the manuscript:

1)    ABWB

2)    SEB

3)    IJMR

4)    LCO

5)    LC-HRMS

6)    UPLC

7)    QTOF

8)    AD

Author Response

Reviewer 2:

Author has performed an interesting research study on the Mennonites population that has been isolated for 25 generations. Author has performed HLA DQ2.5 and DQ8 genotyping along with metabolic profiling approach elucidating high prevalence and altered glutathione pathway. However, there are some serious corrections that need to be rectify in the manuscript.

Major corrections

 Introduction

  • Instead of stating fraction of HLA DQ2.5-positive and/or HLA DQ8 individual, it is advisable to write the proper stats with references.

R.: Thank you for the advice, we specified the required information.

 2)    Line 68-69; There is no need to write (reviewed by)

R.: Excluded as required.

 3)    Page 2, paragraph 3, Line 75-83 must be rewrite; Instead of providing the history of Mennonite population, author is advised to write the genetic background and disease prevalence in relation to celiac disease; number of genetic and epidemiological studies conducted and most important findings in short.

R.: We specified the genetic background and celiac disease prevalence in the first and second paragraphs. Nevertheless we must include the history of the Mennonite population to justify the investigation of this group, in particular.  

4)    Page 2, paragraph 3, Line 82-92; Paragraph seems incomplete. It is advisable to re write this part too depicting the novel objective of the study and the expected outcomes at the end.

R.: We rewrote it as required.

 Materials and Methods

 1)    There are discrepancies in the sample size in more than one section of the manuscript, which arise confusion throughout the study. Author is advised to make a flow chart depicting the number of cases from the enrollment to the number of evaluation of the cases in the study.

 R.: We made a table instead, included in the supplementary material of this manuscript. We also corrected the total number of individuals, which is actually 604 (not 684).

2)    Line 121-123; If the 38 individuals from the pilot study were screened again, then the total number of sample size should be 684+38.

Here author needs to specify clearly whether these 38 individuals were only screened for serology (anti-tTG-IgA or EMA) or interviewed also. And also these individuals were participated in the total sample size or not.

R.: We clarified this as required (secton 22 of Material and Methods).

 3) Author needs to describe the inclusions and exclusions of the study precisely in the manuscript with the basis of proposed sample size.

R.: Done as required. We aimed at 200 individuals from each of the three largest settlements to reach a sample that might be representative of the Mennonite Brazilian population and with enough statistical power to identify an HLA genetic effect (higher than 80%). Inclusion criteria: Mennonite origin of at least one of the parents (sharing a common migratory route from the Netherlands to Poland, then to Ucraine and from there again to Ger-many and later to Brazil or Paraguay), more than 12 years of age, capacity to under-stand and answer the questions of the interviews. Exclusion criteria: desistance to fur-ther participate in the study.

4)    There is no need to provide the irrelevant information in the manuscript. Information related to the current study must provide in the manuscript.

  1. a) Line 98-99; Lactose intolerance survey.

R.: This survey represents the starting point of our investigation. It revealed at first hand an astonishing result of a high prevalence of celiac disease among Mennonites and explains why we undertook this study in this population. We do not deem this information as irrelevant, especially in view of the comment of reviewer 3.

  1. b) Line 104-105; article in local Mennonite newspaper.

R.: We shortened the information.

5)    Line 117-119 repeated second time. It is advisable to shorten the sentence if it contains important information.

R.: This is not simple repetition, since the former section gave information regarding the approach with the Mennonite community itself and its leadership, an information now being required in well-known journals of the epidemiological-clinical interface.

6)    Line 120-121; Reason to conduct the study and the study type must come under the objective section, not in the method section.

R.: Thank you for the suggestion, we moved it to the end of the Introduction.

7)    Line 141-144; Gender, Age and BMI information must come in results section not in method section.

 R.: Moved to the Resuts section, as suggested.

8)    Figure 1: It is advised to the author to create a chart depiction number of cases enrolled from the individual communities, number of serology positive cases from the enrolled cases and the percent of HLA DQ2.5 and DQ8 genotypes.

R.: Done as suggested, but the figure was reallocated to the results section.

9)    Line 147: Serology analysis performed 576 cases. Author has interviewed 684 cases (with 38 cases from pilot study). Author needs to specify what happened to the rest of the cases.

R.: We explained the recruitment and serological screening in details and corrected the total number of interviewed individuals, which is actually 604, in sections 2.2 and 2.3 of Material and Methods.

 10) Line 159-162; Author has mentioned that the serology positive cases were advised to duodenum biopsy based diagnosis of celiac disease, reference 20. Here author needs to specify which diagnostic guidelines were followed for the diagnosis of celiac disease.

R.: We followed the European guidelines (recently updated – Al-Toma A, Volta U, Auricchio R, Castillejo G, Sanders DS, Cellier C, Mulder CJ, Lundin KEA. European Society for the Study of Coeliac Disease (ESsCD) guideline for coeliac disease and other gluten-related disorders. United European Gastroenterol J. 2019 Jun;7(5):583-613. doi: 10.1177/2050640619844125).

 11) Line 165; blood from 400 cases was collected for HLA genotyping. Was there a specific reason to genotype the lesser number of cases? Why the rest cases were left for genotyping?

 R.: Unfortunately, we had financial limitations. Furthermore, we reasoned that a sample size of around 400 (actually 391, we corrected this in section 2.4, in accordance with the information given in Table 1) would be enough to detect a founder HLA effect for celiac disease, given its strong association in European-derived populations, with an estimated power greater than 90%.

12) Author performed HLA DQ2.5 and DQ8 genotyping. Is there any specific reason author did not genotyped HLA DQ2.2 region? Although lesser number of cases show susceptibility with HLA DQ2.2 genotype and has less frequency in celiac disease patients, but still this region has its own importance and it is very important to perform the genetic analysis of this region too. Also, along with DQ2.5, it completes the HLA DQ2 genotype. Author needs to provide very specific reason for that.

R.: Again, we did not genotype this region for financial limitations (this whole project started 2016 with meager funding). Nevertheless, we hope to disclosure the importance of this region in a collaborative study, starting this year. The lack of this information was included among the limitations of the study (6th paragraph of the discussion).

13) Line 179; author described that “we compared 5 CD without GFD”

This statement creating a confusion and require a clarification whether these cases:

  1. a) Newly diagnosed Celiac disease cases
  2. b) Cases from earlier performed pilot study or the cases that were serology positive but did not consuming gluten free diet
  3. c) Cases non-compliant to Celiac disease

R.: They were newly diagnosed CD cases from our study. We included this information in the manuscript.

 14) Line 180-181; author need to write the sample type and sample preparation in short

R.: This information is given in Appendix A.

 15) In the metabolomic analysis section, author needs to write the pathway and the groups of analytes author is targeting and why?

R.: Included in the metabolomic analysis 2.5 section:“ In this study, an untargeted LC-HRMS metabolomics approach was used to com-pare the metabolic state of healthy controls (Mennonites without CD – control group, CT) and Mennonites with CD (serologically positive, newly diagnosed in this study, but not under gluten free diet) to identify differences between their metabolite profiles. This approach was based on the concentration profile of all measurable free low mo-lecular weight metabolites (without targeting a particular analyte/groups of analytes).“.

 16) No need of table 1 and 2 in supplementary sheet.

R.: We prefer to maintain this information in order to ease the access of technical information for reproducibility.

Results

1)     In the table 1, author needs to write the whether the age is in months or years

 R.: Included as suggested.

2)    Did author calculated the frequency of the FDRs and SDRs from the total number of celiac disease cases or from the total number of the cases screened? And why?

R.: Thank you for asking. We calculated the frequencies of how many individuals reported affected FDR and SDR among all interviewed individuals (453). Since this is a genetically isolated population, the share of predisposing genetic variants within the family is probably much higher than in the general admixed Brazilian population, increasing the prevalence of the disease. More importantly, comparing the frequencies among the three different settings allowed us to evaluate the epidemiological differences among them, with Witmarsum being most affected. It furthermore allowed us to get an impression about previous knowledge of the disease in these rather closed communities (of 604, 453 or 75% had previous knowledge about the disease in there families).   We included this in the discussion.   

3)    If all the 10 cases did not underwent biopsy procedures then what guidelines were used to make the diagnosis of celiac disease and how? Line 253

 R.: As mentioned before, we used the European guidelines, recently updated (mentioned in the last paragraph of section 2.3). For ethical reasons, however, we cannot force serologically positive research participants to undertake the endoscopic biopsies. We nevertheless considered them to be positive, given the high specificity of EMA testing. This was discussed as limitations of our study:

R.: Our study has limitations. First of all, we could not confirm the serologically EMA positive results with positive biopsies. However, given the high specificity of EMA testing, false positives are rather unexpected. Second, IgA-deficient individuals may appear negative in our serological screening. Despite being uncommon, they may occur more often among CD patients. Missed CD individuals, in this case, would have increased the prevalence values, already very high. Third, we did not genotype HLA-DQ2.2, which also contributes to CD susceptibility. In a next phase of our genetic-epidemiological investigation, we will screen for all HLA alleles in the Mennonite population. We included this information in the discussion.                                        

4)    Author has previously mentioned that 684 cases were interviewed and out of which 576 samples were tested for serology then why author has calculated the prevalence (serology based and biopsy proven) from 604 cases? From where this 604 figure turn out? Line 255-257

R.: We corrected sample sizes, as shown in Sections 2.2 and 2.3 in Material and Methods.

5)     Why author calculated the frequency of undiagnosed celiac cases from 21 cases and not from the total sample size?

R.: We calculated it among the 21 cases, to have an idea of how many, among all celiac patients in the Mennonite community, miss correct diagnosis (most have complaints). Taking into account the whole sample, it would be 1.66%. This information was included in the text (end of the third paragraph of the Results).

 6)    Unsuspected celiac disease patients. Line 258

Unsuspected is not a standard term, either use atypical or any other specific/appropriate term

R.: We corrected it by using „undiagnosed“ instead.

 7)    In the method section author mention 400 cases were HLA genotyped but here it is written that 391 cases were screened for the HLA variants? Clarify

R.: This was a mistake – the correct number is 391 (corrected in the text).

 8)    Line 282- provide the frequency of HLA genotypes in controls too

R.: We calculated the frequencies of HLA-DQ2.5 and DQ8 carriers in both CD and non-CD participants of the study, please check for them in the last columns and lines of Table 1.

 9)    Previously 21 celiac disease cases were described but in line 290 they are 13 only. Clarify

R.: Only 13 CD patients were among the 391 cases that could be genotyped for HLA-DQ2.5 and DQ8. This information was included in the text.

10) Figure 1 in supplementary sheet must be included in the main text of manuscript.

R.: Included as required.

Discussion

 How did author calculate frequency of underdiagnosed celiac disease cases as 47.6%. why it is not calculated from the total sample size?

R.: We calculated it among the 21 cases, to have an idea of how many, among all celiac patients in the Mennonite community, miss correct diagnosis (most have complaints). Taking into account the whole sample, it would be 1.66%. This information was included in the text (end of the third paragraph of the Results).

2)    Please provide the ratio from the current study as depicted from Finland and Argentina in line 331-332.

R.: Done as requested (almost 1:2, as in Finland).

 3)    Author did not perform the analysis of HLA DQ2.2 region, a well-known predisposing genotype, and stating the possibilities of unknown predisposing genetic variants. Why? Provide hard reasons to support the statement.

R.: Again, we did not genotype this region for financial limitations (please see our former response).

4)    Line 377; please specify the both algorithms in pilot metabolomic investigation.

R.: These are specified in the text, but were added again at the end of the discussion (mummichog and GSEA).

 Minor but important corrections

Author needs to write the full forms of the following at once in the manuscript:

1)    ABWB – excluded from the text

2)    SEB – excluded from the text

3)    IJMR- excluded from the text

4)    LCO - excluded from the text

5)    LC-HRMS – included

6)    UPLC - included

7)    QTOF - included

8)    AD - included

R.: Done as requested.

Reviewer 3 Report

In this original article, Oliveira et al. conducted an epidemiological study in order to assess in a target population (Mennonites) the prevalence of celiac disease (CD). They showed how this population, subjected to particular genetic phenomena, has a particularly high CD rate.

The work requires, in my opinion, some revisions:

Introduction:

1) It should be introduced, when discussing CD in general, what extra-digestive manifestations there may be. Recommended reference.

- https://pubmed.ncbi.nlm.nih.gov/30639642/

- https://pubmed.ncbi.nlm.nih.gov/29933630/

In addition, new evidence has also shown unseen manifestations such as those in the sexual sphere. Recommended reference .(https://pubmed.ncbi.nlm.nih.gov/35419983/).

2) line 64. "who are non-compliant with a GFD and/or have been lately diagnosed develop osteoporosis, infertility, neurological and psychiatric disorders [...]." I would be less "peremptory" and write "may develop" because then it seems that everyone who is not following a GFD indiscriminately develops these complications...

3) I would introduce at the end of the Introduction briefly the purposes of the study. It is important for a reader to have an advance on what the purposes of the study are.

Methods:

It is very interesting to read how the authors described the context in which the patient contacts occurred; it is a good point in the presentation of the study.

1) Section 2.3: From what it seems the sero-screening was done exclusively by assaying anti-transglutaminase IgA but also anti-gliadin. However, the latter were dosed in a population with a median age of 49 years--do you think this is appropriate? Anti-gliadin should have great power especially in pediatric age.... Also, you do not seem to have dosed total IgA..., how can you rule out immunodeficiency in a certain (albeit small) percentage of IgA enrollees? This is a potential bias that would underestimate (albeit by a small amount) the sero-prevalence you calculated. To that end, if you did this should be said in the methods or, if you did not, it should at least be said among the limitations of the study.

2) In addition to what was said before, a positivity for anti-DGP IgA in patients at low risk for CD is predictive for CD in only 15% of cases generating a huge percentage of false positives (https://pubmed.ncbi.nlm.nih.gov/31210940/). I seem to read that you have done duodenal biopsy in all these patients as well? Clarify the point in the "anti-DGP and/or IgA-EmA" methods at line 160 better for this purpose.

Statistics:

The methods are well presented. For significant p-values: clarify whether you have considered, in the tests employed with PRISM or STATA p value two-tailed or one-tailed. This is important to understand the weight of your significance.

Author Response

Reviewer 3

In this original article, Oliveira et al. conducted an epidemiological study in order to assess in a target population (Mennonites) the prevalence of celiac disease (CD). They showed how this population, subjected to particular genetic phenomena, has a particularly high CD rate.

The work requires, in my opinion, some revisions:

Introduction:

1) It should be introduced, when discussing CD in general, what extra-digestive manifestations there may be. Recommended reference.

- https://pubmed.ncbi.nlm.nih.gov/30639642/

- https://pubmed.ncbi.nlm.nih.gov/29933630/

R.: These references were added as suggested. Thank you!

In addition, new evidence has also shown unseen manifestations such as those in the sexual sphere. Recommended reference .(https://pubmed.ncbi.nlm.nih.gov/35419983/).

R.: Highly interesting! Thank you, we added this reference too.

2) line 64. "who are non-compliant with a GFD and/or have been lately diagnosed develop osteoporosis, infertility, neurological and psychiatric disorders [...]." I would be less "peremptory" and write "may develop" because then it seems that everyone who is not following a GFD indiscriminately develops these complications...

R.: Thank you for this remark. We changed the text as suggested.

3) I would introduce at the end of the Introduction briefly the purposes of the study. It is important for a reader to have an advance on what the purposes of the study are.

R.: Done as requested. The introduction ends now with „We performed a pilot study in January 2013, with 93 individuals from the Witmarsum Mennonite community, previously enrolled in a genetic population study for lactose intolerance [21]. We found serologically positive anti-endomysial results in three individuals who unsuspected the disease (3.2%). This prompted us to launch an epidemiological, transversal field study, lasting from October 2016 to December 2018. Daily consumption of gluten, reported in the previous pilot study, further justified a tracking study of celiac disease in this population. In this work, we identified one of the highest CD prevalences worlwide, associated with the distribution of predisposing HLA-DQ variants in Mennonites. Given that the prevalence of CD in first-degree relatives (FDR) of celiac patients in-creases by 2-21%, depending on the population, sex and HLA-DQ genotype [20], pop-ulational screening of the Mennonite population is of high public health value regard-ing preventive medicine, allowing the identification of individuals who will benefit from early dietary intervention, i.e., GFD.“

Methods:

It is very interesting to read how the authors described the context in which the patient contacts occurred; it is a good point in the presentation of the study.

R.: Thank you for the remark. We nevertheless shortened it, following the suggestion of another reviewer.

1) Section 2.3: From what it seems the sero-screening was done exclusively by assaying anti-transglutaminase IgA but also anti-gliadin. However, the latter were dosed in a population with a median age of 49 years--do you think this is appropriate? Anti-gliadin should have great power especially in pediatric age.... Also, you do not seem to have dosed total IgA..., how can you rule out immunodeficiency in a certain (albeit small) percentage of IgA enrollees? This is a potential bias that would underestimate (albeit by a small amount) the sero-prevalence you calculated. To that end, if you did this should be said in the methods or, if you did not, it should at least be said among the limitations of the study.

R.: Thank you for pointing this out. We did not check for IgA deficiencies, and will include these analysis in a further step of our population project, to uncover potential false negatives. We also included this among the limitations of our study, in the discussion.

2) In addition to what was said before, a positivity for anti-DGP IgA in patients at low risk for CD is predictive for CD in only 15% of cases generating a huge percentage of false positives (https://pubmed.ncbi.nlm.nih.gov/31210940/). I seem to read that you have done duodenal biopsy in all these patients as well? Clarify the point in the "anti-DGP and/or IgA-EmA" methods at line 160 better for this purpose.

R.: This is true. Anyway, we confirmed all positive cases with anti-EMA tests. We did also not identify any DGP positive individual, among all screened participants in our study (see supplementary table 1).

Statistics:

The methods are well presented. For significant p-values: clarify whether you have considered, in the tests employed with PRISM or STATA p value two-tailed or one-tailed. This is important to understand the weight of your significance.

R.: We used two-tailed P values. This was clarified in the Methods section (2nd paragraph of section 2.6).

Round 2

Reviewer 2 Report

Dear Author

Thank you very much for providing the revised manuscript.

Reviewer 3 Report

The manuscript was improved